

# Standard source of atmospheric black carbon aerosol generated from ultrasonic spray of BC suspension

Ruchen Zhu[1], Huixiang Wang[1], Xiaoyan Wang[2], Hao Liu[3]

[1]College of Environmental Sciences and Engineering, Peking University, Beijing 100871, China
[2]China National Environmental Monitoring Center, Beijing 100012, China
[3]Yangtze River Water Resources Protection Bureau, Wuhan 430010, China

*Correspondence to: Huixiang Wang (hxwang@pku.edu.cn)*

**Abstract:** Black carbon (BC) aerosol has strong radiative forcing and plays an important role in global climate change and human health. A generator with low levels of BC-air is developed in this study for researchers to calibrate BC
monitors. Ultrasonic nozzle is applied to atomize BC suspension to produce quantificational BC-air samples which can be used directly as a standard source of BC. Membrane test conducted by balance is used to check up its feasibility. Results show that the relationship of weight increment of membrane and target concentration of BC-air have very good linearity. This confirms that the ultrasonic spray system is a good source to generate standard concentration of BC-air. The device has good feasibility in the BC concentrations range of 0-200 μg·m$^{-3}$. Multi Angle Absorption Photometer (MAAP) is
used to detect the concentration of BC-air generated by the ultrasonic spray of suspension. Target concentrations generated by the device accord with the measured data of MAAP.

## 1 Introduction

Black carbon (BC) is a distinct type of carbonaceous material formed from the incomplete combustion of fossil and biomass based fuels under certain conditions (Boucher et al., 2013). It not only plays an important role in the climate, but
also can be a threat to human health. BC is considered to cause warming by reducing aerosol albedo, causing a semi-direct reduction of cloud cover, and reducing cloud particle albedo (Hansen et al., 2000). The effective radiation forcing due to aerosol-radiation interactions of BC aerosol is +0.4 (+0.05 to +0.8) W·m$^{-2}$, and that on snow and ice is +0.04 (+0.02 to +0.09) W·m$^{-2}$ (Myhre et al., 2013). BC absorption in ice and snow can increase globally averaged near-surface temperature and accelerate the melting of snow (Flanner et al., 2007;Jacobson, 2004). Besides, studies have shown that exposure to BC can
result in acceleration of development of atherosclerosis (Niwa et al., 2007;Provost et al., 2016) and in vitro studies ultrafine particles induce a greater oxidative stress than fine particles and this indicates that ultrafine particles may play a role in the toxicological effects (Stone et al., 1998). Given its climate and health effect, it is important to determine the concentration of BC in atmosphere. Standard source of BC for calibration of BC monitors is a vital device. Generally, four methods are used to make BC standard source.



An aerosol generator based on electronic atomization by spark discharge has been successfully applied to produce carbon particles. A small fraction of the electrodes is evaporated by sufficient heat generated from discharge and then the carbon atoms condense to particles (Schwyn et al., 1988). An inert atmosphere such as helium or argon is required to produce pure aerosols (Evans et al., 2003). The size of primary particle formed in this method is about 5 nm and that of agglomerates ranges from 50 to 200 nm. The particle mass flow rate differs from 20 µg·h$^{-1}$ to 7 mg·h$^{-1}$ by simply changing spark frequency (Helsper et al., 1993). It has a great advantage of convenient operation, high reproducibility and producing very clean particles without undesirable by-products (Kuznetsov et al., 2003;Schwyn et al., 1988). One of the drawbacks of this method is that the concentration of BC generated is much higher than the levels in atmosphere. Consequently a great deal volume of dilute gas, such as zero air or nitrogen, is needed during calibration of atmospheric BC monitors with a standard source of BC by spark discharge method.

Incomplete combustion is another way to generate black carbon particles and it includes several burner arrangements such as opposed flow, co-flow, flat flame burners and engines (Stipe et al., 2005). Although particle size distribution depends weakly on operation conditions and type of engine (Mansurov, 2005), engines are not desirable for lab studies of particles because of their great hour and day fluctuations and long break-in-period (Stipe et al., 2005). Methane and other hydrocarbons are used to combust with air to create carbon particles (Kirchstetter and Novakov, 2007;Stipe et al., 2005). The inverted burner system can produce particles from 50 nm to 250 nm and also a large concentration range. The size distribution of produced particles is affected by the fuel and air flow rate and /or by diluting the fuel jet with nitrogen (Stipe et al., 2005). The inverted diffusion flame has the advantages of nearly constant particle production rate and producing particles only composed of BC (Kirchstetter and Novakov, 2007). A simple and low cost combustion aerosol generator can produce particles whose size distribution approximately ranges from 10 nm to several hundred nanometers and particle number concentration is about $10^{14}$ particles per stere (Yawootti et al., 2010). Soot aerosol produced from combustion in a shock tube shows bimodal size distribution at 15-30 nm and 50-100 nm for lean fuel mixtures and 150-700 nm for rich fuel mixtures. The total number concentration of undiluted soot in the shock chamber is $10^5$ to $3×10^6$ particle·cm$^{-3}$ (Khalizov et al., 2012).

Dry powder can also be used to produce particle aerosols. Dry dispersion methods produce particle aerosols by lifting small mass of powder from a rotating source using a pressure drop (Prenni et al., 2000). It includes rotating turntable feeders, rotating scrapers and fluidized bed particle generators (Teague et al., 2005). Strong shear forces are applied in expending flows, mechanical uplifting by a brush, or fluidized bed disintegration (Vlasenko et al., 2005). A fluidized bed dust generator successfully produces particle aerosols with powder feed rate varying from 1.2 to 36 mm$^3$·min$^{-1}$ and the air flow rate from 9 to 30 L·min$^{-1}$ (MARPLE et al., 1978). A modified fluidized bed aerosol generator produces high number densities of $10^5$ particles·cm$^{-3}$ of soot particles with flow rates of below 10 L·min$^{-1}$ (Prenni et al., 2000). The generated particle size distribution is bimodal with one in the supermicrometer diameter size range and the other in the submicrometer diameter size range which is nearly log-normal and whose count median diameter is between 0.1 and 0.3 micrometers





(Prenni et al., 2000). Turntable feeder with a cylinder open at both ends placing on the turntable provides better variation, control and reliability in aerosol output. Dust concentrations range from 100 µg·m⁻³ to 560 mg·m⁻³ with the utilization of two size of the turntable feeder (Reist and Taylor, 2000).

Atomizing liquid solution is another way to generate aerosol particles (Biskos et al., 2008). Pneumatic, ultrasonic, and electrospray atomization are used to produce relatively small primary droplets ranging from 1 to 10 µm in diameter (Biskos et al., 2008). It has a great advantage of producing high particle number concentrations, but the interference of impurities in the solvent is the main limitation of this method (Biskos et al., 2008;Tiwari et al., 2013). An aerosol generator of high stability using syringe pump to provide a constant liquid flow is introduced (Liu and Lee, 1975) and it has been developed and widely used to generate aerosols (Park et al., 2012;Stabile et al., 2013). Vibrating orifice aerosol generator (VOAG) is a kind of ultrasonic nozzle used for generating aerosols (Berglund and Liu, 1973). It has been used to characterize aerodynamic particle sizer (Volckens and Peters, 2005) and time of flight secondary ion mass spectrometry (Palma et al., 2007), to generate aerosols (Lim et al., 2008;Mitchell et al., 1987;Štefancová et al., 2011) and to synthesize particles (Rama Rao et al., 2002). Electrospray produces nearly monodisperse droplets by means of electrical forces and droplets ranging from nanometers to several micrometers can be generated by controlling flow rate of the liquid, the voltage at the capillary nozzle and the electrostatic potential between the liquid and the electrode (Biskos et al., 2008;Jaworek and Sobczyk, 2008). Atomizing a suspension of carbon black in alcohol generates particle concentration of $10^9$ particles·m⁻³ and the size distribution depends on the size of the droplet and suspended carbon particles (Horvath and Gangl, 2003).

## 2 Materials and methods

### 2.1 Reagents

Black carbon powder (particle size ranges from 30-40 nm with an average value of 33 nm); deionized water (Milli-Q Gradient, Millipore); sodium hydroxide (NaOH, analytically pure); absolute alcohol (analytically pure); lauryl sodium sulfate, SDS (analytically pure).

### 2.2 Instruments

Ultrasonic nozzle (8700-120, SONO-TEK, USA); peristaltic pump (BT00-100M, Longer Pump Ltd, China); mass flow controller (007-7B, Sevenstar Electronics Ltd, China); ultrasonic oscillator (DSA50- GL1, Desen Precision Ltd, China); electronic analytical balance (AR1140/C, Ohaus, USA); Multi Angle Absorption Photometer (MAAP)(5012, Thermo Scientific, USA)





## 2.3 Schematic of standard source of black carbon

Figure 1 shows the schematic diagram of the standard source of black carbon:

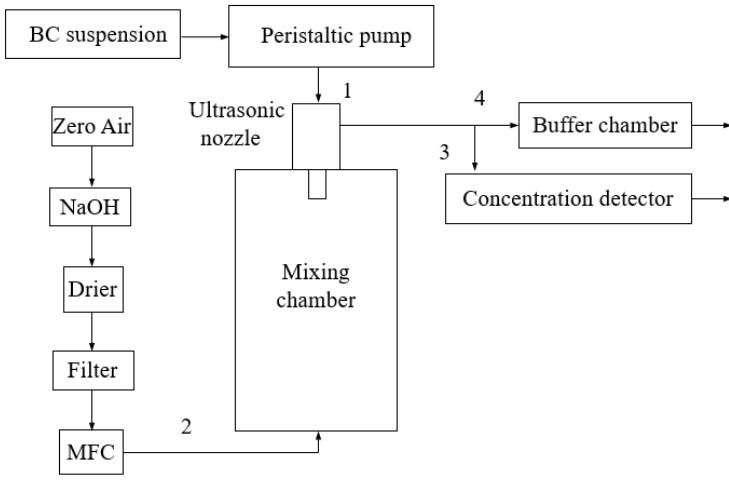

**Figure 1.** Schematic of the standard source of black carbon

Mixing chamber is made of quartzose cylinder, about 2 liters in volume. There are four channels in mixing chamber.

Channel 1: BC suspension enters ultrasonic nozzle through peristaltic pump, and the nozzle atomizes BC suspension to the mixing chamber;

Channel 2: Zero air subsequently passes through absorption tower with NaOH, drier that absorbs acidic gas and moisture, then filter (5 μm) and mass flow controller (MFC), and finally enters mixing chamber;

Channel 3: The concentration of BC in the air is detected by concentration detection system, such as membrane sampling of atmospheric aerosols or MAAP;

Channel 4: The excessive BC-air sample exhausts from this system through a buffer chamber of 200 L.

The concentration of BC in mixing chamber can be changed by adjusting BC suspension concentration, flow rate of suspension controlled by peristaltic pump and zero air flow rate controlled by MFC.

## 15   3 Results and discussion

### 3.1 Preparation of black carbon suspension

The pH scale of black carbon suspension is crucial for BC to suspend in the solution as a whole. Setting a pH of the suspension greater than 9 can stabilize BC so it can suspend for a long period, e.g. pH=9-10. NaOH solution is used to adjust pH of the suspension. Black carbon suspension consists of black carbon, surfactant and solvent. Right mixed ratio of different

solvents and the dosage of surfactant enables BC to suspend for a reasonable period of time in the solution, which ensures BC to be dispersed quantificationally into the air by atomizing.





Water, as well as polar or nonpolar organic solvents can be used to make BC suspension. Alcohol is suitable as a component of BC suspension due to its solvability with water. Experiments have confirmed that BC suspensions at pH=9-10 with mixed ratios 1:0, 4:1, 1:1 and 1:4 of absolute alcohol to deionized water and surface active agent (SDS) showed good stability. In order to reduce the amount of $H_2O$ that atomizing into mixing chamber, alcohol and deionized water mixed with the ratio of 4:1 is

used as solvent.

The ratio of BC and SDS is determined by experiments. Prepare 5 beakers with BC suspension, each one is 250 mL and includes 5.0 mg BC and solvent of alcohol and deionized water (4:1). Add 0.5, 1.0, 2.5, 5.0 and 10.0 mg SDS into the 5 beakers respectively and the ratios of SDS to BC are 1:10, 1:5, 1:2, 1:1 and 2:1. The suspensions are shaken by ultrasonic for 30 minutes and set at rest for 24 h. The two suspensions with ratios 1:1 and 2:1 of SDS to BC are stable and deposition of BC

appears in other 3 beakers. The ratio 1:1 of SDS to BC is chosen for BC suspension.

### 3.2 Working power of ultrasonic nozzle

The ultrasonic nozzle is 8700-120 from SONO-TEK and it can provide liquid flow from 0.024 to 24 mL·min$^{-1}$ which meets the requirements of BC standard source.

The working power of ultrasonic nozzle should be kept in a certain range to generate ideal spray because there would be not

enough energy for atomization if it is too low and there would be large water mass instead of spray if it is too high. According to its user manual, 1 W is chosen as the best working power in our study.

### 3.3 Calibration of peristaltic pump

Both deionized water and suspension of 4 mg·L$^{-1}$ BC are used to calibrate peristaltic pump. When peristaltic pump is running, roll extrusion stress of feeding tube remains unchanged. Liquid is collected for more than 20 h and flow rate is determined

by balance and timer. The correlation between liquid flow rate and rpm of the pump shows very good linearity. For suspension of 4 mg·L$^{-1}$ BC, the linear correlation coefficient $R^2$ equals to 0.9998. It also suggests that the stability of suspension meets the demand as a black carbon source.

As additives, atomized amount of suspension should be kept in a low range in order to make sure that the humidity of BC-air sample is not too high, so liquid flow of 50 or 100 μL·min$^{-1}$ is chosen and the corresponding rotate speed is 2.6 or 5.2

r·min$^{-1}$.

### 3.4 Feasibility test of BC standard source by membrane sampling

The BC-air sample formed in mixing chamber passes through sampling membrane in channel 3 under a certain flow and sampling time. The mass difference of the membrane is weighed up by balance and the actual mass of BC in BC-air sample is obtained. The gradient analysis of actual BC mass obtained from membrane under different target concentrations of BC-

air sample is conducted to inspect whether the source can produce BC-air sample with a certain concentration gradient range.





Blank experiments using zero air and solvent (alcohol to water = 4:1) are carried out to examine two kinds of membrane, JN (nylon, pore size 0.15 μm, dia. Φ47 mm) and PTFE (Polytetrafluoroethylene, pore size 0.22 μm, dia. Φ47 mm).

Zero air passes through the two kinds of membrane individually in 10 L·min⁻¹ for 24 h. No observable weight increment is observed for both kinds of membrane.

Both kinds of membrane are dipped in solvent of alcohol to water, 4:1, for 24 h, then air them in 50 ℃ for 2 h. Also no observable weight increment is observed for both kinds of membrane.

Both kinds of membrane sample spray generated by ultrasonic nozzle for 24 h individually, which is composed of zero air (10 L·min⁻¹) and solvent (only alcohol and water in 4:1), then air them as above, JN membrane has no observable weight increment and the mass of PTFE membrane increases obviously.

Blank experiments confirm that hydrophilic JN membrane is suitable for test of BC standard source by membrane sampling.

According to the liquid flow of peristaltic pump and sampling time, the theoretical BC mass of spray is obtained. The correlation of theoretical mass increment and actual mass increment of membrane is analysed to see the feasibility of BC standard source. Table 1 shows the parameters of membrane sampling test.

**Table 1.** Membrane sampling test parameters

| Items | Values |
| --- | --- |
| Target concentration of BC-air sample /$C_t$ | 20~100 μg·m⁻³ |
| Flow of Zero air /$V$ | 10 L·min⁻¹ |
| Concentration of BC suspension /$\omega$ | 4~20 mg·L⁻¹ |
| Rotate speed of peristaltic pump/$r$ | 2.6~5.2 r·min⁻¹ |
| Liquid flow of ultrasonic nozzle /$u$ | 50~100 μL·min⁻¹ |
| Sampling time /$t$ | 20 h |

Theoretical concentration of BC-air sample $C$ is calculated by Eq. (1)

$$C = \omega u/V \qquad\qquad (1)$$

Theoretical weight increment of membrane after 20 h sampling, $\Delta m_T$, is calculated by Eq. (2)

$$\Delta m_T = \omega u t \qquad\qquad (2)$$

Actual weight increment of membrane after 20 h sampling, $\Delta m_A$, is obtained by balance.

Figure 2 shows membrane mass difference, $\Delta m_T$, and $\Delta m_A$, in different target BC-air sample concentration when $u$ is 50 μL·min⁻¹.





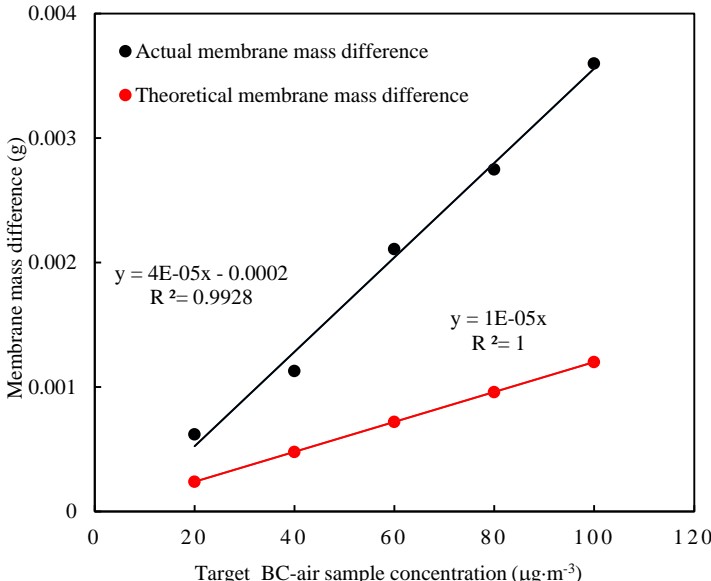

**Figure 2.** Membrane mass difference under different target BC-air concentrations

Actual membrane mass difference shows obvious linear relationship with BC-air sample concentration. It indicates that this ultrasonic spray of suspension can serve as standard source of BC. But actual differences are much higher than

theoretical ones. The deviation from the theoretical values is from the component, SDS and NaOH in the solution. Actual weight increment of membrane, $\Delta m_A$, results from three parts: BC, SDS and NaOH.

$$\Delta m_A = \Delta m_{BC} + \Delta m_{SDS} + \Delta m_{NaOH} \qquad (3)$$

$\Delta m_{BC}$, $\Delta m_{SDS}$ and $\Delta m_{NaOH}$ are weight increment of membrane owing to BC, SDS and NaOH individually. Supposing total BC, SDS and NaOH in the suspension can be filtrated by membrane, expressions are deduced: $\Delta m_{BC} = \Delta m_{SDS} = \omega u t$ because of

the ratio of BC to SDS is 1:1 and $\Delta m_{NaOH} = 0.4 u t$ at pH 9. Take $V= 10$ L·min$^{-1}$ and $u=100$ μL·min$^{-1}$, thus following equation is obtained

$$\Delta m_A / t = 0.02C + 0.04 \qquad (4)$$

Equation (4) shows that the relationship of $\Delta m_A / t$ and C is linearity. To confirm the linear relationship between $\Delta m_A / t$ and $C$, experiments are carried out on membrane sampling of BC suspension. Table 2 shows the data.

**Table 2.** Sampling data of BC suspension by membrane

| Concentration of BC in suspension /$\omega$ (mg·L$^{-1}$) | Sampling time /$t$ (min) | Increment of membrane /$\Delta m_A$ (μg) | Theoretical concentration of BC in air /$C$ (μg·m$^{-3}$) |
|---|---|---|---|
| 5 | 689 | 600 | 41.4 |




| | | | |
|---|---|---|---|
| 5 | 687 | 500 | 34.6 |
| 7.5 | 663 | 800 | 57.4 |
| 7.5 | 714 | 900 | 60.0 |
| 10 | 672 | 1200 | 85.0 |
| 10 | 704 | 1300 | 87.9 |
| 15 | 731 | 1600 | 104.2 |
| 15 | 782 | 1700 | 103.5 |
| 20 | 714 | 2100 | 140.1 |
| 20 | 726 | 2100 | 137.8 |

Figure 3 shows relationship of $\Delta m_A / t$ and $C$ using data in Table 2. It has a very good linearity, $R^2=1$. This confirms that the ultrasonic spray system is a good source to generate standard concentration of BC-air. By varying concentration of BC in suspension, changeable concentration of BC in air can be achieved using the system.

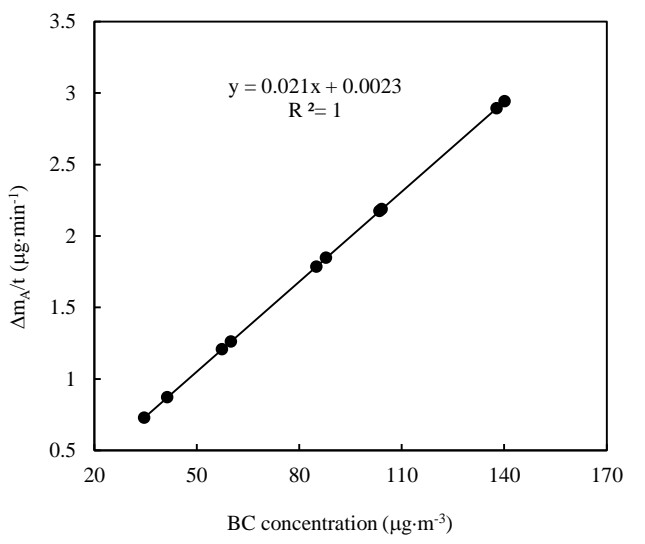

**Figure 3.** Relationship of $\Delta m_A / t$ and $C$ derived from data in Table 2

The slope and intercept of line in Fig. 3 are derived by linear regression with data in Table 2. As compared with Eq. (4), both slopes in Eq. (4) and in Fig.3 almost are the same. But intercepts are very different. Take notice of the process obtaining Eq. (4), it transpires the first term on the right of equal sign of Eq. (4) owes to SDS and the last term to NaOH. These suggest almost all SDS in suspension is filtrated out by membrane but NaOH is not.

The precision of the balance used in our study is 0.1 mg, and it is a little poor when weighing the membrane differences which have a precision of 10 μg, but it has little influence on linear relationship of target BC-air concentrations and membrane mass differences. Balance with higher precision can be applied in our further study to reduce the weighing error.



## 3.5 Test of BC-air standard source by MAAP

MAAP measures aerosol light absorption at three detection angles from synchronous measurements of radiation passing through and scattering back from a fiber filter where particles are collected (Petzold and Schönlinner, 2004). This method needs no data correction for scattering effects and no parallel-measured aerosol light-scattering coefficients (Petzold et al., 2005;Petzold and Schönlinner, 2004). It is used to test the BC standard source. Target BC-air concentrations are set as 0.5, 1.0, 3.0, 5.0, 10.0, 20.0 µg·m$^{-3}$ by varying concentration of BC suspension. Parameters of MAAP test are shown in Table 3.

**Table 3.** MAAP test parameters

| Items | Values |
| --- | --- |
| Target concentration of BC-air/$C_t$ | 0.5~20.0 µg·m$^{-3}$ |
| Flow of zero air/$V$ | 10 L·min$^{-1}$ |
| Extraction flow of MAAP | 500 L·h$^{-1}$ |
| Concentration of BC suspension/$\omega$ | 0.05~4 mg·L$^{-1}$ |
| Rotate speed of peristaltic pump/$r$ | 2.6~5.2 r·min$^{-1}$ |
| Liquid flow of ultrasonic nozzle/$u$ | 50~100 µL·min$^{-1}$ |
| Sampling time/$t$ | 25 min |
| Time resolution/$R$ | 1 min |

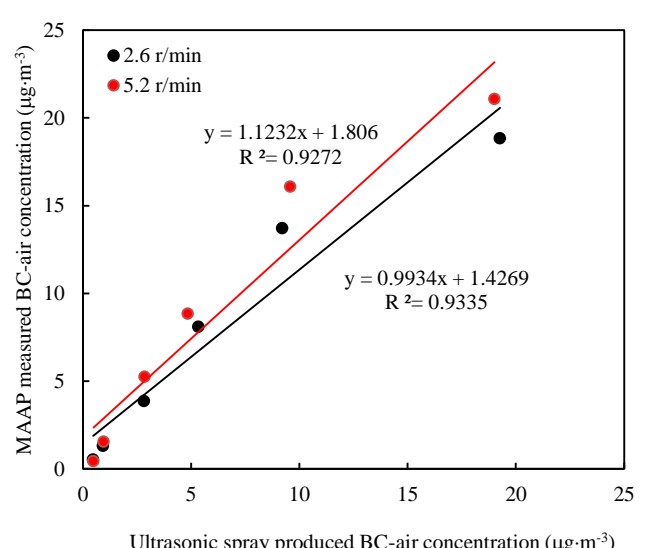

**Figure 4.** MAAP measured and ultrasonic spray produced BC-air concentration with 2.6 and 5.2 r·min$^{-1}$ rotate speed of peristaltic pump



MAAP measured and ultrasonic spray produced BC-air concentrations show good linearity with target concentration of 0.5~20 μg·m⁻³ when rotate speed of peristaltic pump is both 2.6 and 5.2 r·min⁻¹ in Fig. 4. It indicates that the self-designed standard source has high feasibility in this concentration range.

The solvent of BC suspension contains water, and it may increase the humidity of BC-air sample. When liquid flow of
ultrasonic nozzle is 100 μL·min⁻¹, the concentration of water imported by solvent can be calculated as 2 mg·L⁻¹, corresponding to the relative humidity <12 % at ambient temperature, which has little influence as a source of atmospheric black carbon aerosol.

Theoretically, the lower limit of BC-air concentrations produced by ultrasonic spray is 0, but in our study the value is 0.5 μg·m⁻³ because of the detection limit of BC monitor. The upper limit concentration is limited by the stability of concentration
of BC suspension. When a stable BC suspension is 20 mg·L⁻¹, the concentration of BC-air samples produced by this system is 200 μg·m⁻³.

## 4 Conclusion

In this study, we develop a generator of BC aerosols by atomizing BC suspension of certain concentrations with ultrasonic nozzle to produce low levels of BC-air aerosol which can be used directly as a standard source to calibrate BC monitors.
Membrane test shows that this system has good feasibility in the BC-air concentration of 20~100 μg·m⁻³ and MAAP test shows that this system also has good feasibility in the low BC concentration range of 0.5~20 μg·m⁻³.

## Competing interests

The authors declare that they have no conflict of interest.

## Acknowledgements

This work was financially supported by National Natural Science Foundation of China (41475111).

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
