# Peer review of "Standard source of atmospheric black carbon aerosol generated from ultrasonic spray of BC suspension"

_Atmospheric Measurement Techniques, 2017_

## Referee Comment (RC1) · Anonymous Referee #1 · 5 Oct 2017

General comment

The manuscript aims to present a freshly developed generator of black carbon suitable for calibration of BC-measuring instruments especially at low concentrations. The authors introduce the variety of nebulisation techniques, describes the design of the generator and at the end its performances. The here presented technique is based on the use of a commercially available ultrasonic nozzle and does not present any particular technical innovation in the field of aerosol nebulisation for instrument calibration or BC quantification in rain-snow samples. Beside this, the structure of the paper is confused and several parts of the text need rewriting. For these reasons I do not recommend the publication of the present article. The specific criticisms are addressed as following. Considering that the work presented here needs a general rethinking, editing errors were ignored.

Main points

1. Motivations. The major criticism points to the motivations and goals at the base of this work. However the authors give an overview of different nebulisation approaches, is not clear what is the technology gap the authors want to fill and in which way this gap affects the calibration of BC-measuring instruments. Thus, due to the fact that motivations were not explicitly stated, the final goals of the manuscript are not clear. For example, the quantification of BC in snow samples is usually based on liquid nebulisation by concentric pneumatic nebulisers (Katich et al., 2017; Lim et al., 2014; Mori et al., 2016; Wendl et al., 2014), the ultrasonic nebulisation might represent an alternative. Naturally, this must be asserted showing current limitations of pneumatic nebulisation, advantages of ultrasonic approach and supported by a systematic comparison. I encourage the authors to reconsider their motivations and goals, redesign their generator, and run a complete series of new test.

2. Technical description. This is a technical manuscript, as consequence the reader expect a consistent amount of technical details. The present manuscript lacks in information, some descriptions are incomplete, and the technical drawing appears to be wrong.

3. The generator and its performances. Without considering the main criticism described above, the manuscript does not present any results of scientific interest. First, the aerosol generator, it operating principle, technical details are poorly described. The key element of the generator, the ultrasonic nozzle, is poorly described and its advantages compared to other techniques remain unexplained. Second, the treatment of the aerosol after nebulisation is not explained. The nozzle produces mist of fine water drops containing the BC particles. The authors should explain how the water is

removed from the sample air (heating-cooling stages, diffusion dryer) and how many particles might be lost. One of the main problems of the entire paper is that the nebulisation and transport of the aerosol is considered to be 100% efficient. Third, the gravimetric method used is not appropriate, out-dated and affected by large uncertainty. See specific points. The main limitation is represented by its un-specificity to black carbon: everything that is nebulised together with BC can contribute to the change in weight of the membranes. As said before, the calculations done by the authors rely on the fact that nebulisation efficiency is of 100% and that the milliQ water does not contain particles.

4. Data treatment. From the graphs and description it appears that the paper is based on a limited number of observations. The authors should state how many experiments and repetitions were performed for each configuration in order to asses repeatability and uncertainty. The latter is never assessed and no error bar are shown in any graphs, limiting the consistency of the results.

5. Conclusions. Is not clear what the authors want to demonstrate and which were the achievements. A poor conclusive chapter reflects this. Generally, the conclusions of a manuscript represent the climax of a work, where authors summarise the major finding of the presented work, improvements compared to previous works, impacts of their research, eventual limitations and, possibly, future applications or improvements. Without considering the scientific relevance of the presented findings, the conclusions of this manuscript are superficial and not developed in broader contest.

Specific points

P3/L18 The introduction gives a nice and complete overview of the existing techniques for aerosol nebulisation, but the motivations and goals of the present study are not explained. The author should provide them in the introduction. P3/L21 What type of black carbon soot was used? P3/L22 I imagine that with "absolute alcohol" the authors referrer to ethylic alcohol (ethanol). I suggest using the chemical nomenclature. The

full name of SDS is specified in page 5 line 3. Specify the full name here or remove the acronym from this list. P3/L25-27 According to the title, the ultrasonic nozzle is the key element of the present work. However a brief and general description of ultrasonic generation of spray is given in the introduction, no technical description is provided in the methodology section. Short but exhaustive description of the operating principle, technical limitations (flow rate) and performances (mean droplet size) must be presented. Similar action is required for the MAAP. Basic information on the balance precision and measuring limits should be given as well. The membranes are only described at page 6. The description should be moved in the technical section. P4/L5-14 Figure 1 is here described. First I would suggest changing the name of "zero air", this is a bit miss leading. In case of non-operating nozzle, this can act as zero air. During liquid injection and nebulisation, the airflow acts as purge air (or sampling air in case of membrane sampling). The description does not match the figure diagram: from the picture it appears that channel 3 and 4 are linked to the ultrasonic nozzle and not the mixing chamber. Please correct the diagram accordingly. Nebulization of liquids creates a mist: suspension of liquid droplets in the air. Most part of commercial nebulisers (Marin-5, CETAC; APEX nebuliser) is equipped with warming and cooling stages in order to dry the sample flow. At each cooling step, the condensing water is removed by mean of peristaltic pumps. The authors should explain how the condensing water is extracted from the chamber, what's the RH at the output of the chamber and how and if the outgoing air flow is dried. Quantification of absorption by filter based absorption photometers is in fact sensitive to relative humidity levels. P4/L16 The authors explain that high pH stabilises BC. They are asked to specify the meaning of "stabilize" and the principle behind it. P5/L2-3 Reference needed. Again the use of "stability" is generic and not clear. P5/L7-10 Here the effect of SDS on BC suspensions is described. The effect is to decrease deposition effects, supportive data must be provided in the text or in the supplementary material by mean of the MAAP or membranes. It looks like the procedure adopted to prepare the suspension is wrong. The authors describes that SDS is added to a suspension of 250 ml, which already contains the BC powder. If

this is the case, this is wrong. Solution must be brought to the desired volume after addition of SDS. P5/L11-16 This subchapter belongs to the method section. P5/L17-25 Proportionality between the rpm of a peristaltic pump and the provided liquid flow is always expected. Due to the fact that the peristaltic pump is a commercial version, I do not see the need of this section. If the authors think that this information is needed, I suggest presenting a figure, possibly in the supplementary material. The choice of a low liquid flow was made in order to keep low the RH in the mixing chamber. However, liquid flow also control the BC concentration. Moreover, the RH in the mixing chamber is not only regulated by the liquid flow, but by the "zero air" flow, the volume of the tank and the temperature. If the authors decide to maintain the present section, all the argumentations must be supported by data and graphs: RH variability in the mixing chamber, pump calibration curves, effect of temperature. The use of diffusion drier will most probably help reducing the RH in channel 3. P6/L3-10 No data evidence. P7/L18 This is true if the nebulisation efficiency is 1 and there are zero losses in the mixing chamber and in the tubing. Considering that each BC particle is imbedded in a liquid droplet having a diameter of about 18 um (manual of the nozzle) and no heating is applied to the mixing chamber I strongly doubt that no particles are lost from nebulisation to detection. The author should estimate the transmission efficiency of solid and liquid particles troughs their system. P7-L7 This is true when the sampling flow does not contain any water. Are the membrane dried before weighting? Nebulizing liquid solutions of NaOH, ethanol and SDS can experimentally support the validity of the equation and quantify the mass increase not due to BC presence on the filter. The authors must consider that milliQ water contains large amount of small particles, which can positively bias the here presented calculation. Previous works have shown that the nebulisation efficiency of aerosol generators is never of 100%. Here the authors assumes that every single particle of BC and SDS is nebulised and transported to the filter without losses. P8/5-9 I do not understand what the authors try to show. Captions of figures and tables are usually not part of the text. P8/L10-12 Balance performances must be reported in the technical section and not here. The precision of the balance

seems to be poor for performing any kind of mass quantification, but the relative uncertainty is not reported. An example from Table 2: for the first case the mass increment of the filter is of 0.6 mg, the error introduced by balance precision is of 17%. No uncertainty or standard deviation is shown, what is the repeatability of the measurements? How many times the weighting experiments have been repeated? P9/L2-5 However the MAAP is a well-known instrument and fully described in other papers, some more details might be given. The authors should specify what MAC was used to convert absorption coefficient to BC mass concentration. Please state the wavelength according to (Müller et al., 2011). MAAP requires a complex data correction, but it is simply implemented in the instrument software. P10/L1 From figure 4 the proportionality between MAAP eBC and theoretical BC mass is not linear. Additionally, the difference in eBC mass between 2.6 and 5.2 RPM changes with BC liquid concentration. At low BC concentrations, the observed eBC is smaller at 5.2 RPM than 2.6 RPM. Can the author explain this?

---

## Short Comment (SC1) · 16 Oct 2017

1. Motivations. Our research aims to develop a BC aerosol generator of low concentrations without large amount of dilute gas such as nitrogen. The concentrations of BC aerosols produced by this generator can be as low as 0.5 ïА■g/m3, and the theoretical lower limit of the generator we developed is 0. 2. Technical description. Technical information and descriptions can be further added in the revised paper. 3. The generator and its performance. First, the operating principle, technical details and other information can be added in the revised edition as said above. Second, the calculated relative humidity of the aerosol sample is below 12% at ambient temperature, which is

much small than atmospheric humidity and has little influence on the aerosol generated so we didn't remove the water from the sample. Third, the membrane test can show that nebulization efficiency is nearly 100%, error can be further calculated. 4. Data treatment. We have done many experiments and the repeatability is very good. Not all data are shown in this article due to the limited thesis length. 5. Conclusions. This part will be rewritten in the revised article to demonstrate the motivations, achievement, limitations and also future developments.

---

## Referee Comment (RC2) · Anonymous Referee #2 · 28 Nov 2017

Summary:

In this work, the authors use an ultrasonic nozzle to generate black carbon aerosol (BC) from a pH-controlled, surfactant-containing, water-alcohol suspension. An aerosol stream from the ultrasonic nozzle was mixed with zero air in a 2-L chamber and then collected on nylon and Teflon filters for 20-h or analyzed with a Multi-Angle Absorption Photometer (MAAP) after 25 minutes. Mass difference between aerosol-laden filters and blank filters exhibit a linear relationship with a theoretically-derived BC concentration. Interestingly, the mass experiment membrane mass difference is higher than the theoretical mass difference. MAAP results are also linear, but similarly higher than the

theoretical mass difference.

As currently written, this paper exhibits a clear lack of substantial results and also contains excessive formal deficiencies. The foremost of these are outlined below, with the reviewer's suggestions to address them. As addressing comments may cause a major re-working of the paper, only major comments are included in this work.

Major Comments:

1. Currently, this manuscript does not fit within the scope of Atmospheric Measurement Techniques (AMT). Manuscripts in AMT report new developments, significant advances, or novel aspects of laboratory measurement techniques. As written, it is unclear if this technique is novel and, if so, what are its advantages over the four methods summarized in the introduction of the paper. The reviewer is especially interested in its advantages over discharge generators and inverted-burners, which make black carbon in-situ and without any additional water or surfactants. Similarly, what is the advantage of this particular setup over other methods that atomize liquid solutions described in the last paragraph of the introduction?

2. Both the mass-difference and MAAP results are above their theoretical estimations. This suggests that an additional calibration is needed with this technique to get quantitative results. The reviewer would guess that the experimental results would be below the theoretical estimations due to losses in the mixing-chamber and experimental setup. Do the authors have an explanation as to why both techniques give higher results? Consequently, do the authors also have a recommendation for the field on how this technique can be used "directly as a standard source" given that the results do not quantitatively match their theoretical estimations?

3. Given that the two above major comments can be addressed, the reviewer suggests that the authors pay special attention to correct word choice and grammar in the next revision. For example, in the abstract alone, the reviewer notes that the phrase "check up its feasibility" is grammatically incorrect, and the phrase "good feasibility in

the BC concentrations range" is grammatically vague due to word choice. This persists throughout the paper. Furthermore, the paper contains several instances where incomplete sentences are intentionally used. These include the Reagents and Instruments sections, Table Headings, and Figure Captions. While it may be appropriate in other fields to use incomplete sentences in these instances, it is generally not commonplace in Atmospheric Science journals; the reviewer suggests that the authors amend these sections.
* * *